# Comparison of Detomidine or Romifidine in Combination with Morphine for Standing Magnetic Resonance Imaging in Horses

**DOI:** 10.3390/vetsci11030124

**Published:** 2024-03-08

**Authors:** Cecilia Vullo, Enrico Gugliandolo, Vito Biondi, Marco Biffarella, Giuseppe Catone, Adolfo Maria Tambella

**Affiliations:** 1Department of Chemical, Biological, Pharmaceutical and Environmental Sciences, University of Messina, Viale Ferdinando Stagno D’Alcontres, 31, 98166 Messina, Italy; 2Department of Veterinary Sciences, University of Messina, Polo SS. Annunziata, 98169 Messina, Italy; enrico.gugliandolo@unime.it (E.G.); vito.biondi@unime.it (V.B.); giuseppe.catone@unime.it (G.C.); 3Independent Researcher, Via Anna Salamone, 7, 98073 Mistretta, Italy; marco.biffarella@gmail.com; 4School of Biosciences and Veterinary Medicine, University of Camerino, 62024 Matelica, Italy; adolfomaria.tambella@unicam.it

**Keywords:** horses, standing magnetic resonance imaging, constant rate infusion, sedation depth, ataxia

## Abstract

**Simple Summary:**

In equine patients, when possible, specific procedures are preferably performed under deep sedation to decrease the risk of mortality and morbidity associated with general anesthesia in these animals; however, the sedation protocols used must be effective and not dangerous, that is, they must not induce respiratory depression or ataxia, which are essential features of both diagnostic and surgical procedures. A constant rate infusion of different alpha-2 agonists and opioids is recommended to induce a balanced and stable level of sedation and analgesia during a surgical procedure and to avoid a possibly higher risk of over- or undersedation with a repeated single bolus. Recently, standing magnetic resonance imaging has been introduced as an advanced diagnostic equine imaging technique. During the procedure, the complete immobility of the horse is of crucial importance in order to avoid the acquisition of poor-quality and non-diagnostic images. The goal of this prospective, blind clinical trial was to compare the quality of sedation, the presence/absence of ataxia, and the effects on the respiratory and cardiac systems caused by the administration of an infusion of romifidine or detomidine with morphine at a constant rate in standing horses subjected to a low-field magnetic resonance imaging examination. This study also aimed to determine which of the two sedative protocols is more suitable for avoiding patient motion and the risk of motion artifacts.

**Abstract:**

The aim of this study was to determine the most appropriate sedation protocol for a standing magnetic resonance imaging (MRI) examination in horses, comparing continuous rate infusions (CRIs) of detomidine and romifidine combined with a single bolus of morphine. Sixteen horses referred for standing low-field open-magnet MRI were randomly assigned to one of two sedation protocols. The horses were premedicated with 0.03 mg/kg of intramuscular acepromazine, and those animals belonging to Group D received an intravenous (IV) loading dose of detomidine (0.01 mg/kg) 30 min later, while those of Group R received romifidine (0.04 mg/kg). If the horses were inadequately sedated, an additional dose of IV detomidine (0.005 mg/kg) or romifidine (0.02 mg/kg) was administered, according to the animal’s group. During the MRI, a single IV bolus of morphine (0.05 mg/kg) was administered, and according to which group it belonged to, the animal started the administration of detomidine (0.01 mg/kg/h) or romifidine (0.02 mg/kg/h). Heart rate (HR), respiratory rate (RR), rectal temperature (RT), depth of sedation, and degree of ataxia were evaluated every 10 min during MRI. Two horses belonging to Group D and four horses from Group R needed additional sedation before entering the MRI unit because they were unsatisfactorily sedated. No side effects were observed following morphine bolus administration. During the MRI procedure, five horses in Group R received an additional IV romifidine bolus (0.01 mg/kg) because the depth of sedation score was 1 and the ataxia score was 0. Any substantial differences were recorded between the two treatments in terms of HR, RR, and RT. In conclusion, at the doses used, a detomidine–morphine combination following a CRI of detomidine appears more suitable than a romifidine–morphine combination following a CRI of romifidine for maintaining an adequate depth of sedation and adequate immobility in horses undergoing standing MRI.

## 1. Introduction

Thus far, magnetic resonance imaging (MRI) has been an effective diagnostic tool in equine lameness examination, especially in evaluating the distal extremities [1]. Recently, standing low-field MRI has become available for use in horses, eliminating the complications related to general anesthesia [2,3].

However, the physiologic advantages of standing sedation are counterbalanced by the intrinsic complication of maintaining appropriate patient restraint [4]. To perform a surgical or diagnostic standing procedure in horses, alpha-2 adrenergic receptor agonists alone or in combination with opioids and/or ketamine are used as a single bolus, multiple boluses, or a loading dose followed by a constant rate infusion (CRI), with the aim of providing more balanced and steady-state sedation and reducing adverse effects, episodes of excessive or inadequate sedation, and ataxia [4,5,6,7,8,9].

During most standing procedures, a minimum amount of movement by the horse may be accepted, unless it compromises the horse, the operator’s safety, or the procedure itself. The scenario is different during standing MRI focusing on the limbs, where the complete immobility of the horse is required in order to avoid the risk of the motion artifact and to obtain the best-quality images [3,4]. Another challenge is the long procedure of the MRI, with acquisition times of approximately 90 min.

Neuroleptanalgesia is an anesthetic technique that consists of a combination of drugs with different pharmacologic actions, with the aim of reducing the doses of single drugs, and thus decreasing their adverse effects. The final result is a balanced anesthesia, such as that achieved by a combination of alpha-2 adrenoreceptor agonists and opioids [10].

Morphine is a mu opioid receptor agonist that has been used alone for analgesia and with sedatives to facilitate standing surgery [11,12,13]. The level of analgesia provided increases with the dose of morphine but so do the adverse effects [12].

Alpha-2 adrenoreceptor agonists are the most commonly used drugs in horses for standing sedation. For standing procedures requiring long-term chemical restraint, as is the case for MRI, a CRI of these drugs is often administered [4,14,15].

Detomidine and romifidine are alpha-2 adrenergic drugs, widely used in clinical equine practice, that provide sedation, analgesia, and muscle relaxation. They have several pharmacologic effects typical for this group of drugs and are characterized by a reluctance to move, reduced responsiveness to environmental stimuli, bradycardia, peripheral vasoconstriction, increased arterial blood pressure, decreased respiratory rate, decreased intestinal motility, and increased diuresis. The degree and duration of sedation are dose-dependent [16,17,18,19].

Many studies have compared these two drugs using bolus administration and a CRI, in both experimental and clinical studies, showing them to be potent analgesics and sedatives [13,15,16]. However, to the authors’ knowledge, the feasibility of either of these protocols in standing sedation for MRI has not been investigated yet. Therefore, the aim of this study was to determine whether detomidine or romifidine CRI, combined with a single bolus of morphine, provided adequate sedation and immobility with minimal cardiorespiratory effects in horses undergoing standing MRIs.

## 2. Materials and Methods

The study protocol, designed according to the Good Scientific Practice Guidelines and adhering to European legislation, EU Directive 2010/63, was approved by the Veterinary Sciences Department Ethics Committee of the University of Messina (approval number 016/2023). After the informed consent of the owner, sixteen adult jumping horses (twelve females and four males), with an age of 5–14 years, a body weight ranging between 430 and 560 kg, and American Society of Anesthesiologists (ASA) physical status of 1 or 2 that were referred to the Veterinary Teaching Hospital of Messina University for standing MRI were enrolled in this study. Horses with cardiac dysrhythmias, impaired respiratory function, or liver disorders; in a pregnant or lactating state; or with behavioral modifications (e.g., aggression) impacting safety were excluded from this study. The owners’ written consent was obtained for each animal.

Horses were fasted for 8 h prior to MRI with water ad libitum. Basal rectal temperature (RT, °C), heart rate (HR, beats/min), and respiratory rate (RR, breaths/min) were recorded before any drug administration.

An intravenous (IV) catheter (14 gouge × 16 cm) was inserted into one of the jugular veins following hair clipping, skin disinfection, and subcutaneous local lidocaine (Lidocaine 2%, Fatro, Ozzano dell’Emilia, Italy). Horses were randomly assigned to one of two treatment groups (Group D or Group R, composed of eight animals each) by using the random number generator, GraphPad QuickCalcs software (GraphPad Prism 10 for MacOS, version 10.1.1, GraphPad Software Inc., San Diego, CA, USA).

The same anesthetist (C.V.) that evaluated the animals was blinded to the administered treatment, which was prepared and injected by an assistant (VB). All horses were premedicated in a stable with 0.03 mg/kg of intramuscular (IM) acepromazine (Prequillan, Fatro, Italy), and 30 min later, they received a 0.01 mg/kg intravenous (IV) loading dose of detomidine (Dorum, Acme, Vicenza, Italy) (Group D) or 0.04 mg/kg of romifidine (Sedivet, Boehringer Ingelheim, Milan, Italy) (Group R) near the MRI room.

After 10 min, the degree of sedation was evaluated by using a 4-point sedation scale [20] (see Table 1). Horses that were deemed inadequately sedated (score 0 or 1) before entering the MRI unit received a supplementary dose of detomidine (0.005 mg/kg) or romifidine (0.02 mg/kg) according to the animals’ group.

Before placing the animals for MRI examination, a single IV bolus of 0.05 mg/kg of morphine (Morfina cloridrato, Molteni, Giussano, Italy), diluted in 0.9% NaCl (Saline solution, S.A.L.F., Bergamo, Italy) to a volume of 10 mL, was administered, and after 5 min, detomidine (0.005 mg/kg/h) or romifidine (0.01 mg/kg/h) IV infusion was started by using an infusion pump (B Braun, Melzungen, Germany, Vista Basic Infusion Pump) whose VB was adjusted up or down following the decision of C.V., who assessed the depth of sedation and degree of ataxia. The infusion did not exceed 0.01 mg/kg/h of detomidine or 0.02 mg/kg/h of romifidine. The intravenous solutions of alpha-2 agonists were prepared by adding 25 mg of detomidine or 20 mg of romifidine to a 500 mL bag of saline to obtain a concentration of 0.05 mg/mL for both drugs. Just before the start of the MRI scan, a urinary catheter providing a urine collection system was inserted into the female horses, while a bucket was placed near the penises of the male horses, and cotton was placed in the ears to decrease the response to auditory stimuli. HR (with palpation of the mandibular pulse), RR (by observing thoracic excursion), RT (by using a digital thermometer), sedation depth, and ataxia degree were evaluated as soon as the infusion started and every 10 min by using a 4-point scale [21] (Table 2). A supplementary bolus of detomidine (0.005 mg/kg) or romifidine (0.01 mg/kg) was administered when the horses were inadequately sedated (sedation score 0 or 1 and ataxia score 0) in Group D or Group R, respectively, despite the maximum administered infusion dosage.

During the procedure, a support stand was placed in front of the horses to support their heads and stabilize the animals.

Evaluation of gut motility was evaluated with the auscultation of the four quadrants every 30 min for the next 6 h, using a score of 0–3 (0—silent, no motility heard for 30 s; 1—less than normal motility; 2—normal motility; 3—hypermotility, more gut sounds than usual).

Cardinal data were assessed for normality of the data distribution by using the Shapiro–Wilk test and reported as the mean ± sem (standard error of the mean). The comparisons between groups were assessed with an unpaired t-test. Longitudinal comparisons within each group were performed by using the repeated measures ANOVA and the Holm–Sidak post hoc test. Ordinal data were reported with median and min–max range. The Mann–Whitney test and the Friedman test with Dunn’s multiple comparison test were used to compare the ordinal variables between groups and within each group individually. The frequencies of horses requiring an additive dose of sedative were analyzed with Fisher’s exact tests.

Differences between the two groups were considered statistically significant with a *p*-value <0.05. GraphPad Prism 9 for MacOS, version 9.3.1 (GraphPad Software Inc., San Diego, CA, USA) was used for the analysis.

## 3. Results

The population under study did not show significant differences between groups for age (mean years ± sem, Group D: 8.75 ± 0.90; Group R: 9.12 ± 1.14; *p* = 0.8002), weight (mean kgs ± sem, Group D: 497.5 ± 20.62; Group R: 476.9 ± 12.85; *p* = 0.4101), or sex (Group D: two males and six females; Group R: two males and six females).

There was no difference between groups regarding the basal parameters of HR (beats per minute, Group D: 36.5 ± 0.5; Group R: 35.1 ± 0.6; *p* = 0.1123), RR (respiratory acts per minute, Group D: 13.5 ± 0.5; Group R: 13.5 ± 0.5; *p* > 0.99), or RT (°C, Group D: 37.2 ± 0.1; Group R: 37.2 ± 0.1; *p* = 0.8903).

All horses underwent an MRI examination because of forelimb navicular syndrome. In all animals, only one leg was investigated.

An MRI exam was completed in all animals, although the exam in one horse in Group D was postponed until the next day because the animal appeared too nervous when it had already been in the MRI room, probably due to a panicked reaction. The examination was performed 3 days later, allowing the owner to enter the MRI room in order to make the animal feel more comfortable.

Regarding the sedation score evaluated before MRI, in Group D, 0.01 mg/kg of IV detomidine was considered suitable (score 2) in six out of eight animals. Two animals showed an inadequate sedation score and needed an extra dose of 0.005 mg/kg of IV detomidine before entering the MRI box: one horse had a particularly nervous temperament (score 0) and the other horse showed fear near the MRI box (score 1). Concerning Group R, the depth of sedation with 0.04 mg/kg of IV romifidine was considered adequate (score 2) in four out of eight animals: the other four animals in Group R needed an extra dose of 0.02 mg/kg of IV romifidine before entering the MRI box, because the depth of sedation was considered unsatisfactory (score 1). Nevertheless, no significant difference emerged from the comparison in all animals for the frequencies of animals requiring an additional bolus of sedation before MRI (Group D: 2 horses; Group R: 4 horses; *p* = 0.6084).

The sedation score before MRI was found not to be significantly different between groups (Group D, median 2, range 0–2; Group R, median 1.5, range 1–2) (Figure 1).

The time between sedation and the administration of the morphine bolus was 7 ± 3 min.

A significant decrease in HR, RR, and RT (*p* < 0.05) was observed when comparing the baseline values and each time point during MRI within each of the two groups. The trends of HR and RR remained rather constant during MRI, with clinically irrelevant differences, and there were no statistical differences between groups at any time point (*p* > 0.05) or any longitudinal differences within groups considering HR in Group D (F = 0.9347, *p* = 0.4956); HR in Group R (F = 0.6313, *p* = 0.7481); RR in Group D (F = 1.703, *p* = 0.1180); and RR in Group R (F = 0.5089, *p* = 0.8446) (Figure 2A,B). The administration of morphine triggered a transitory increase in HR in just one horse in Group D five minutes after detomidine bolus administration.

The RT values showed a gradual decreasing trend during MRI, both in Group D (F = 35.87, *p* < 0.0001) and in Group R (F = 42.88, *p* < 0.0001), but without showing any difference between groups at any time point (*p* > 0.05) (Figure 2C).

During MRI, no significant differences between groups or within each group were found for the depth of sedation and ataxia (Figure 3).

However, in Group R, three animals received a supplementary bolus of romifidine (0.01 mg/kg) because they showed a score of 1 for the depth of sedation and a score of 0 for ataxia (two animals at 30 and 40 min, and one animal at 80 min, respectively), despite the CRI of romifidine being increased to 0.01 mg/kg/h. In Group D, no horses required an increase in the CRI or additional sedation, since two animals scored 1 for the depth of sedation score (one horse at 50, 70, and 80 min, and one horse at 60 min), but no animals scored 0 for ataxia score, as shown in Figure 3. Thus, compared to group D (no horses), a significant number of horses in group R (five horses) needed an extra dose (*p* = 0.0256) (Figure 4).

The duration of MRI showed significant differences between groups (t = 2.248; *p* = 0.0412) and was longer in Group R (96.38 ± 2.909 min) than in Group D (87.88 ± 2.416 min).

The first intestinal auscultation was performed 30 min after the end of the MRI examination. A score of 2 was assigned to all animals at 30 min, 1, 3, and 6 h.

## 4. Discussion

During long-term standing procedures for horses, alpha-2 agonists are commonly used as a single bolus followed by a CRI [4,5,6]. Several studies have shown that combining alpha-2 agonists with opioids increases the level of sedation and analgesia, improves chemical restraint, and decreases the response of the animal to environmental stimulation [4,7,8,9]. Two studies have demonstrated that the combination of methadone and the infusion rate of detomidine improves sedation and analgesia in horses [14,22].

In this study, the authors investigated the depth of sedation, degree of ataxia, HR, RR, and RT by using a combination of morphine with two different CRI protocols with detomidine or romifidine in horses undergoing MRI examination, based on previous similar studies.

In this study, the dosages used for both the initial bolus and the CRIs of detomidine and romifidine were similar to or slightly lower than those reported in other studies [16,17,23], considering the minor pain stimuli during the MRI procedure, due to maintaining the animals’ full weight throughout the duration of the study. This may be the reason why more top-ups were needed in Group R. Although Evrard et al. used a bolus of detomidine (0.01 mg/kg) and morphine (0.1 mg/kg), followed by a continuous drip of morphine (0.05 mg/kg/h) and detomidine (0.022 mg/kg/h) in NaCl (0.9%), in horses undergoing standing MRI examinations [24], in this study, a lower dosage of a CRI was chosen as the authors are more familiar with the rate used in this study. Furthermore, considering the side effects observed with dosages equal to or higher than 0.1 of morphine (increased locomotion, decreased gastrointestinal borborygmi, increased heart rate) [25], a dosage of 0.05 mg/kg of morphine was used.

Solano et al. assessed the behavioral and cardiorespiratory effects of a combination of 5 μg/kg of medetomidine and 0.05 mg/kg of morphine, followed by a CRI of medetomidine (5 μg/kg/h) and morphine (0.03 mg/kg/h), in seven horses undergoing standing exploratory laparoscopy. They concluded that the protocol resulted in adequate sedation and stable cardiorespiratory function [11]. In our study, the same low IV dose of morphine (0.05 mg/kg) was used as a single dose, without an added CRI of morphine, because we performed a diagnostic exam that did not include pain related to surgical stimulation. In another study, behavioral changes were noted following 0.2 mg/kg of morphine administration, which included sweating and muscle fasciculations, and flared nostrils, muscle tremors, and ataxia occurred after a bolus of 0.5 mg/kg of morphine; behavioral modifications were not manifested after the administration of either 0.05 or 0.1 mg/kg of morphine [26]. Similar results were obtained in the present study, where none of the side effects mentioned above occurred after a bolus of 0.05 μg/kg of IV morphine. In addition, we used the co-administration of acepromazine, which seems to minimize excitement-related potential side effects [27,28,29]. In fact, it has been demonstrated that acepromazine, due to its antidopaminergic effect, blocks the locomotor effects of fentanyl and morphine in horses [27].

The antinociceptive, cardiorespiratory, and sedative effects of a medetomidine constant rate infusion with morphine, ketamine, or both were recently investigated in a horse [9], demonstrating that a CRI administration of 5 μg/kg of medetomidine combined with 50 μg/kg of morphine is safe and provides a proper degree of antinociception and sedation, in the absence of clinically relevant changes in cardiorespiratory variables. Similar results were obtained in this study, where the range of HR and RR remained constant in both groups, with the exception of one horse in Group D that presented tachycardia 5 min after the end of morphine inoculation. In both groups, the RR decrease after the administration of morphine was probably related to morphine-mediated respiratory depression. In addition, although an IV injection of morphine was associated with swaying, sweating, muscle twitches or tremors, and body jerking [12,25], we did not observe any of these side effects in our study.

The RT values progressively decreased in all animals, probably due to the low temperature present in the MRI unit, which is necessary for the proper function of the instrumentation. Although the trend of the lowering of RT was statistically significant in both groups, it cannot be considered clinically relevant, since the maximum lowering compared to the baseline value was 0.75 °C in Group D and 0.57 °C in Group R.

Similarly, as demonstrated, 0.046 mg/kg of romifidine may limit ataxia and the lowering of the head in sedated horses compared to 0.013 mg/kg of detomidine [15]; this study showed superposable results, as all animals in Group R showed an ataxia score between 0 and 1 in comparison with Group D, which showed an ataxia score of 1 throughout the examination.

The levels of sedation and ataxia scored in this study, although apparently better in Group D, did not result in a significant difference. However, more horses required additional sedation in Group R (4 of 8) than in Group D (2 of 8) before starting MRI, although there was no significant difference. During MRI, the difference between groups was so different that it became statistically significant. Five out of eight horses required an additional bolus of the sedative during MRI in Group R, while no horses required it in Group D. Group D, therefore, showed a level of sedation comparable to Group R but was probably more stable over time, allowing the diagnostic exam to be completed with greater safety.

Opioids, when used alone or administered quickly, are often characterized by side effects such as increased HR [26]. To avoid this, in this study, morphine was given slowly and diluted in 10 mL of saline solution. Moreover, morphine exerts a constipation effect in normal horses, as in other species, because it reduces propulsive gastrointestinal motility [12,30]. For this reason, the intestinal motility of the horses included in this study was monitored using the auscultation of gut sounds in each abdominal quadrant for at least 30 s.

The statistical difference in the duration of MRI between groups is related to the depth of sedation shown by four subjects in Group R. In fact, the scan was stopped twice for three subjects and three times for one subject to correctly replace the animals.

The limitations that may have affected the results of this study include the following: (1) A greater number of enrolled animals would have been necessary to confirm the obtained results and increase the inferential power of this study, although the same sample size is also reported in other studies [31]. (2) The scoring system used to assess the depth of sedation and the degree of ataxia during MRI [11] may be replaced with a simpler and faster system, such as the head height above the ground (HHAG) system [32,33]. (3) During MRI, neither the CRI rate nor the total drug dose used were evaluated. (4) The loading dose of romifidine was lower than that in other publications. This is because, in this study, we supposed that the addition of morphine would decrease the need for the initial dose of romifidine. Nevertheless, four animals in Group R needed an extra dose of 0.02 mg/kg of IV romifidine before entering the MRI box, because the depth of sedation was considered unsatisfactory. In comparison, detomidine might have been run at a high rate in all horses from the beginning, making them more stable.

## 5. Conclusions

Although both protocols showed similar sedation and ataxia scores and are feasible without major cardiorespiratory side effects, the results of the current study suggest that a detomidine CRI, at the doses used, combined with a single bolus of morphine, provides a more stable condition for sedation than a romifidine CRI in horses during standing MRI.

In group D, unlike in group R, which required more top-ups, it was possible to complete the MRI examination without the need to extend the time of the diagnostic procedure or administer additional boluses of the sedative. Group R showed a lower degree of ataxia, although it was not statistically significant; this result will prompt further investigations with higher doses of romifidine to obtain deeper sedation.

## Figures and Tables

**Figure 1 vetsci-11-00124-f001:**
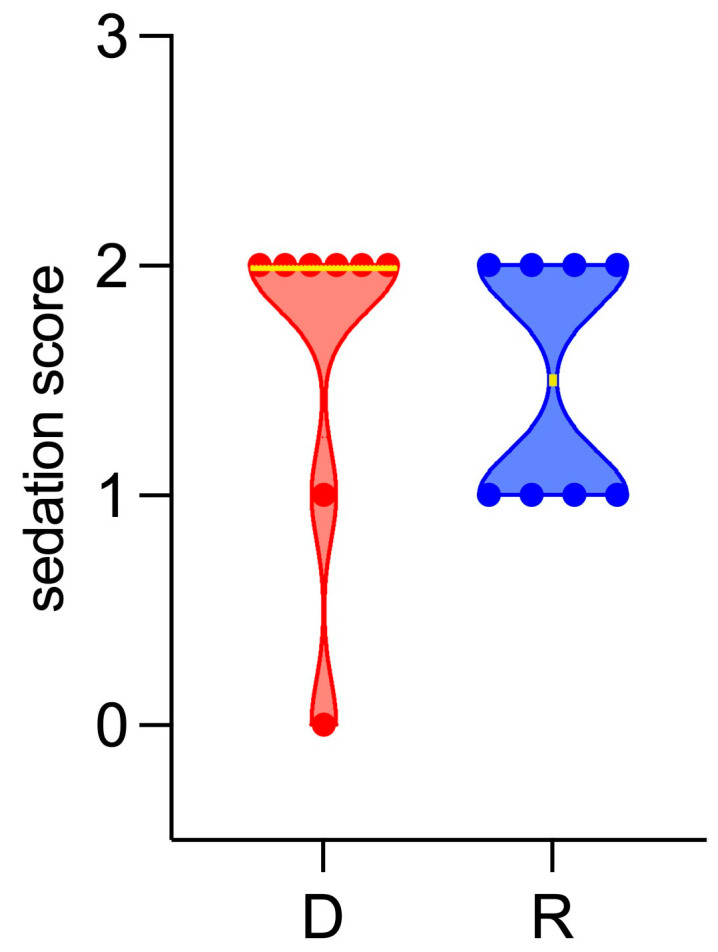
Violin plot showing sedation score before MRI. D: Group D (detomidine), in red; R: Group R (romifidine), in blue. Yellow lines indicate the median. Dots indicate the single measurements scattered in the two groups.

**Figure 2 vetsci-11-00124-f002:**
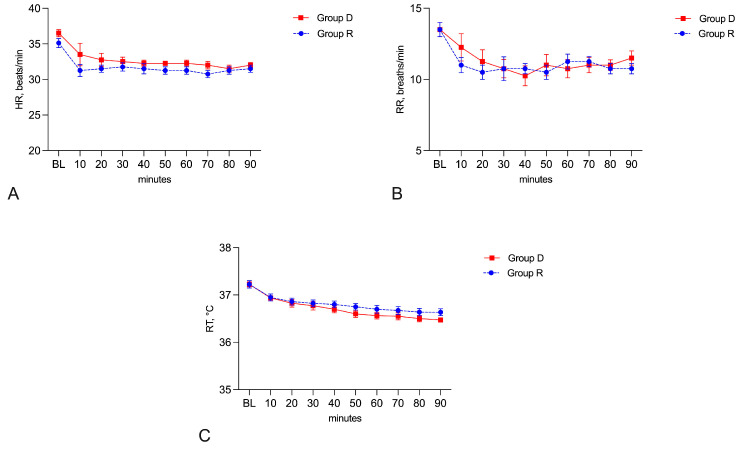
Heart rate (**A**), respiratory rate (**B**), and rectal temperature (**C**) found at baseline (BL) and during MRI in the two groups. Data are presented as mean ± sem (standard error of the mean).

**Figure 3 vetsci-11-00124-f003:**
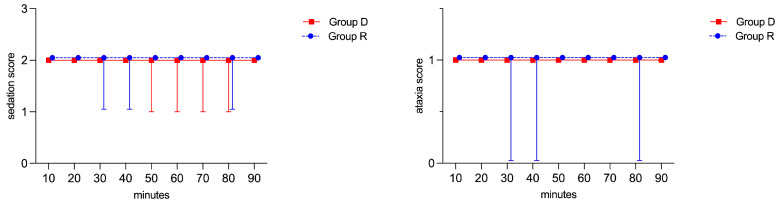
Depth of sedation score and ataxia score during MRI in the two groups. Both scoring systems range from 0 to 3. Data are presented as median and range.

**Figure 4 vetsci-11-00124-f004:**
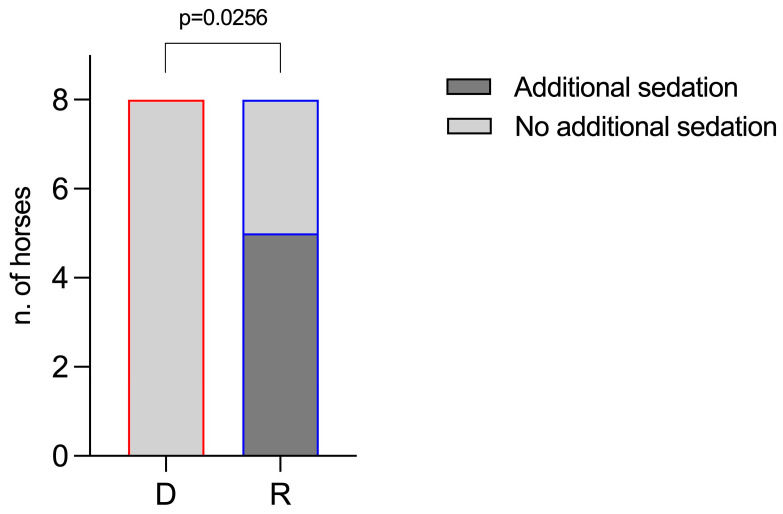
Frequency of horses requiring or not requiring an additional dose of sedative during MRI in the two groups. D: Group D (detomidine), with red border; R: Group R (romifidine), with blue border. The dark-gray-filled bars indicate the number of horses requiring additional sedation; the light-gray-filled bars indicate the number of horses that did not need it.

**Table 1 vetsci-11-00124-t001:** Depth of sedation scoring system (according to Vullo et al., 2017 [20]).

Scores	Definition of Scoring
Poor: 0	(Fully responsive to environment, lips apposed, no lowering of the head, no drooping of the ears)
Mild: 1	(Still responsive to environment, slight separation of the lower lip, slight lowering of the head, slight drooping of the ears)
Good: 2	(No response to environment, separation of the lower lip, lowering of the head, drooping of the ears)
Heavy: 3	(No response to environment, extreme lip separation, pronounced loss of postural tone and ataxia, pronounced separation of the ears tips)

**Table 2 vetsci-11-00124-t002:** Assessment of depth of sedation and degree of ataxia scores used during MRI (from Schauvliege et al., 2019 [21]).

Scores	Definition of Scoring
**Depth of Sedation**	
Score 0:	(No sedation. Animal is alert with normal posture and response to environment/contact with assessor. Normal objection to intervention. Ears responsive to surroundings and moving).
Score 1:	(Mild sedation. May or may not lean on head support; relaxed facial muscles. Reduced responses to background activity in the room. Ears partially responsive to surroundings. Light or no ptosis of the ears).
Score 2:	(Good sedation. Leans on head support. No response to background activity in the room. Pendulous lower lip. Ears mildly responsive to surroundings. Moderate ear ptosis. Eyelids partially closed).
Score 3:	(Marked sedation. Leans strongly on head support. No response to background activity in the room. Pendulous lower lip. Pronounced ear. Ptosis; minimal/no movement of ears. Eyelids partially or fully closed. Eye may be rotated; little to no movements of the eye).
**Ataxia**	
Score 0:	(Standing square; bearing equal weight on all four legs).
Score 1:	(One hind limb in resting position and/or slight swaying).
Score 2:	(Clear swaying or leaning: not maximally bearing weight on one of the four limbs).
Score 3:	(Very pronounced leaning: possibly not bearing weight on several limbs- and/or attempts to become recumbent).

## Data Availability

Data are contained within the article.

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
