# Peer review of "Comparison of Detomidine or Romifidine in Combination with Morphine for Standing Magnetic Resonance Imaging in Horses"

_vetsci, 2024, doi:10.3390/vetsci11030124_

Round 1

Reviewer 1 Report

Comments and Suggestions for Authors

Dear authors,

In this paper you compare two different anesthetic protocols to be used for horses undergoing MRI. The study is very interesting. I have some questions and some suggestions that you can find below. 

I also suggest moderate editing of the English language: I’ve noticed that there are some errors; I have addressed only some of them.

Regards

Introduction

Line 51: in evaluating the distal extremities

Lines 56-58: These are probably the complications you are referring to in line 55. Try to combine these two sentences.

Line 66: change reduces to “to reduce”

Materials and methods

Line 104: why did you include 16 horses?

Line 109: remove “were noted”

Lines 113-114: for a better comprension order the parameters of the sentence (HR was measured by auscultation, RR by observing thoracic excursion and RT was measured using a digital thermometer.) in the same order of the sentence above.

Line 116: change “clipped” into “clipping”

Lines 116-117: provide the volume, the concentration and the trade of lidocaine used

Lines 120-121: “The animals were sedated and evaluated by the same anaesthetist unaware of alfa2-agonists administered”. The same anaesthetist evaluating the animals was blinded to the treatment administered. Please provide some details explaining how did you assure that the anesthetist was not aware of the treatment administered.

Line 128: Did you evaluate again the sedation after the additional bolus?

Line 132: how many minutes after the sedation was the morphine administered?

Lines 134-135: how many minutes after the bolus was the infusion administered?

Lines 140-141: “HR, RR, RT, depth of sedation and degree of ataxia were assessed every 10 min using a 4-point Schauvliege et al. scale”. I guess that all these parameters were assessed every 10 min, not only the ataxia. Please correct the sentence.

Also, did you start the monitoring 10 min after commencing the infusion or 10 min after the sedation or 10 min after morphine? Please specify

Lines 145:146: How did you record the variation in the intestinal motility? Did you use a score? Did you compare the results between the two groups?

Lines 149-158: in the statistical session indicate how do you report the data in the text

Results

Lines 162-163: “and the indication for MRI examination (forelimb navicular syndrome)” change into: all horses underwent MRI examination because of forelimb navicular syndrome.

Lines 168-175: please provide the average sedation score (median or mean depending on the distribution). 

Figure 1: consider if this figure is necessary since the difference is not statistically significant. 

Line 219: please provide also the mean length of the procedure even if it is already represented in the figure.

Line 224: Did you quantify the urine production? Report the results and I also suggest comparing the results between the two groups.

Discussion 

Line 239: remove the dot before the square bracket

Lines 278-283: Merge this discussion with the consideration previously reported concerning medetomidine and morphine at line 247.

Line 273: I suggest writing in an impersonal manner e.g.: similar results were obtained in the present study

Lines 295-296: Isn’t this consideration in contrast to what you state before at line 282-285? Also, please merge this sentence with the discussion above concerning the ataxia induced by detomidine or romifidine (lines 282-285). 

Lines 297-299: is the same discussion of lines 280-281. Please put together these two sentences

The discussions are mainly focused on the effect of morphine. I suggest to improve the discussion section with consideration on the statistical differences that you observed in term of sedation and on the differences between the detomidine and romifidine. 

Also please explain why did you chose the dosages reported here (improve the consideration at lines 304-307) and provide some comparison with previous studies. More recent refernces on alpha2 agonists can be provided.

Lines 312-313: this conclusion is not supported by the results: see lines 201-202 (“During the MRI, no significant differences between groups (p>0.05), nor within each group, were found in the scoring system used to assess depth of sedation and ataxia). The conclusion must be reformulated

Lines 314-315: can you objectively describe the consideration concerning the image quality? Please move this consideration into the discussion section and not in the conclusion

Comments on the Quality of English Language

I suggest a moderate editing of the English language

Author Response

Dear Reviewer,

thank you very much for your suitable and constructive comments. As suggested by you, we tried to improve the quality of the manuscript on these bases, attempting to bridge the limitations you observed.

Best regards 

Cecilia Vullo and co-authors

Introduction

Line 51: in evaluating the distal extremities
Done

Lines 56-58: These are probably the complications you are referring to in line 55. Try to combine these two sentences. 
Thank you, I modified as suggested

Line 66: change reduces to “to reduce”
Done

Materials and methods
Line 104: why did you include 16 horses?
Right observation. We have included only 16 horses in the stage because still the demands to perform MRI are not many

Line 109: remove “were noted”
Ok, done

Lines 113-114: for a better comprehension order the parameters of the sentence (HR was measured by auscultation, RR by observing thoracic excursion and RT was measured using a digital thermometer.) in the same order of the sentence above.
I deleted the period because it was similar to another work.

Line 116: change “clipped” into “clipping
ok, done

Lines 116-117: provide the volume, the concentration and the trade of lidocaine used
Thank you, done.

Lines 120-121: “The animals were sedated and evaluated by the same anaesthetist unaware of alfa2-agonists administered”. The same anaesthetist evaluating the animals was blinded to the treatment administered. Please provide some details explaining how did you assure that the anesthetist was not aware of the treatment administered.
Thank you for the suggestion. I reformulated the phrase and I added that the treatment was prepared by an assistant.

Line 128: Did you evaluate again the sedation after the additional bolus?
No, we didn’t.

Line 132: how many minutes after the sedation was the morphine administered?
This time has not been calculated, but approximately it has been no less than 10 minutes

Lines 134-135: how many minutes after the bolus was the infusion administered?
 I added that the infusion started 5 minutes after morphine administration

Lines 140-141: “HR, RR, RT, depth of sedation and degree of ataxia were assessed every 10 min using a 4-point Schauvliege et al. scale”. I guess that all these parameters were assessed every 10 min, not only the ataxia. Please correct the sentence.
Thank you. I corrected as suggested modifying assessed to evaluated.

Also, did you start the monitoring 10 min after commencing the infusion or 10 min after the sedation or 10 min after morphine? Please specify
Thank you for the suggestion. I specified that HR, RR, RT, depth of sedation and degree of ataxia were evaluated as soon as the infusion started and every 10 min…….

Lines 145:146: How did you record the variation in the intestinal motility? Did you use a score? Did you compare the results between the two groups?
The auscultation was performed with a stethoscope for 3 minutes in the right paralumbar fossa, and bowel sound was assigned a score of 0-3 (Sasaki, N., Murata, A., Lee, I. and Yamada, H., Res. Vet. Sci. 84, 305-310, 2008) but we did not specify this because a difference between the two groups, although it would have been an added value, was not made in this study.

Lines 149-158: in the statistical session indicate how do you report the data in the text
Thanks, done. Cardinal data were reported as mean and sem; Ordinal data were reported as median and range. 

Results
Lines 162-163: “and the indication for MRI examination (forelimb navicular syndrome)” change into: all horses underwent MRI examination because of forelimb navicular syndrome.
Ok, done.

Lines 168-175: please provide the average sedation score (median or mean depending on the distribution). 
Median and range of sedation score before MRI in each of the two groups was reported. In the previous version of the manuscript the authors had not reported the median sedation score and range to avoid repetition, given that they believed that such data could be easily extrapolated from the violin plot.

Figure 1: consider if this figure is necessary since the difference is not statistically significant. 
Figure 1 has been deleted

Line 219: please provide also the mean length of the procedure even if it is already represented in the figure.
Thanks. The length of the procedure was added for both groups. 

Line 224: Did you quantify the urine production? Report the results and I also suggest comparing the results between the two groups.
I’m sorry but we didn’t quantify the urine production so we didn’t compare the different between groups.

Discussion 

Line 239: remove the dot before the square bracket
Ok, thanks.

Lines 278-283: Merge this discussion with the consideration previously reported concerning medetomidine and morphine at line 247.
Sorry, but I didn’t understand.

Line 273: I suggest writing in an impersonal manner e.g.: similar results were obtained in the present study
Ok, done

Lines 295-296: Isn’t this consideration in contrast to what you state before at line 282-285? Also, please merge this sentence with the discussion above concerning the ataxia induced by detomidine or romifidine (lines 282-285). 
Right, I replaced ataxia with depth of sedation

Lines 297-299: is the same discussion of lines 280-281. Please put together these two sentences
Thank you. The two sentences were merged. 

The discussions are mainly focused on the effect of morphine. I suggest to improve the discussion section with consideration on the statistical differences that you observed in term of sedation and on the differences between the detomidine and romifidine. 
Thank you for the suggestion. Discussion was implemented with considerations on statistical differences observed in sedation and about differences between detomidine and romifidine.

Also please explain why did you chose the dosages reported here (improve the consideration at lines 304-307) and provide some comparison with previous studies. More recent references on alpha2 agonists can be provided.
Ok, done.

Lines 312-313: this conclusion is not supported by the results: see lines 201-202 (“During the MRI, no significant differences between groups (p>0.05), nor within each group, were found in the scoring system used to assess depth of sedation and ataxia). The conclusion must be reformulated

Lines 314-315: can you objectively describe the consideration concerning the image quality? Please move this consideration into the discussion section and not in the conclusion.
Ok, done

Comments on the Quality of English Language
I suggest a moderate editing of the English language
Ok, we provided to send the paper to improve English editing.

Reviewer 2 Report

Comments and Suggestions for Authors

The authors present an interesting manuscript that, while not particularly novel, could have some relevance as standing MRI units become more widespread in practice. I am afraid that I don't really understand the presentation of the data or the conclusions, however. In the introduction, the authors state that they want to compare 2 CRI protocols, then in the methods they state that a variable rate infusion was used (with a range of doses), then nothing more is said about the infusions - what was the final dosing?, how much total was needed? - was one adjusted more than the other? - etc. It seems that there is a lot of information missing on this aspect of the research. In the conclusion, the authors state that detomidine was likely a better choice than romifidine, but this is based on data (better images) that were not actually evaluated. Based on what you have presented - all the exams were completed, the ataxia and sedation levels were not different, etc. - it seems like both drugs are quite equal. If the authors feel like detomidine presents an advantage in this setting as far as image quality, I would suggest having a few radiologists look at all the images and score them based on some quality rubric and then compare that between groups.

Other specific comments:

Can the title be a bit more specific - what exactly were you evaluating (not just the drugs used)? Also there seems to be an extra 'a' in the title

Line 53 - What does the incidence of complications in horses undergoing GA have to do with your study? If you want to say that GA is associated with more complications that standing sedation for MRI, please explain this better.

Line 70 - Is there really a need for complete immobility during ultrasound (and perhaps x-ray) examinations? The ultrasound probe tends to move with the patient, does it not?

Line 73 - I'm not sure what multimodal analgesia has to do with standing MRI procedures? Do you need analgesia for non-invasive imaging?

Line 93 - As noted above, your entire aim was to compare a CRI of the 2 drugs, but there is very little mention of CRIs in the rest of the manuscript.

Line 97 - Your hypothesis has nothing to do with your aim, and wasn't even evaluated in the study. Since every horse received morphine, how could you evaluate whether the 'addition of morphine' did anything? Some of the horses would have needed to not receive morphine in order to evaluate whether the addition of it was beneficial or not.

Line 106 - Does it matter to your results what brand of MRI it was? Please remove if it doesn't.

Line 108 - How did you evaluate the patients for 'impaired respiratory or liver function'?

Line 116 - What is a 'bottom of lidocaine'?

Line 134 - Was the infusion dose of drugs varied during the procedure? Or was the infusion rate constant, but different for each horse? Either was, the dosing needs a little more explanation.

Line 136 - Your math on the concentrations here seems a little off - if you put 25mg of detomidine in 500mL of saline, the concentration is 0.05mg/mL, not 0.8mg/mL.

Line 150 - Since you performed multiple comparisons on individual patients, you should include the effect of the individual horse as a random factor in your modeling - and I'm pretty sure that Prism cannot handle random factors in this way. Please check this with your statistician and note how the effect of the individual horse was handled.

Line 170 - You explain the reason for additional sedation in the detomidine group, please do this for the romifidine group or simply note that 2 and 4 horses needed additional sedation in their respective groups.

Figures 1 & 5 are redundant to what you noted in the text. Please remove.

In Figure 4, you display mean and SEM, but since these scales are ordinal you  should probably show median and range (none of the horses scored a sedation of 1.8...).

Line 195 - So the slope of the line was not zero, were there any differences within groups when comparing different time points? eg. was the RT at 90 minutes significantly different than that at 0 minutes? Also, in the graph your baseline RT (and other physiologic variables) is not shown - you should add this.

Line 209 - If all animals in group D had a sedation score of 'up to 1' (meaning less than 1), shouldn't they all have received additional boluses of sedation?

Figure 6 could probably just be noted in the text as average duration and SD or SEM.

How did you know bowel motility resumed 20 minutes after the end of the exam when you were only evaluating every 30 minutes? Also, what does the 'slowly' modifier mean here?

Line 257 - What exactly were you evaluating that might indicate behavioral changes?

Line 282 - So detomidine causes more ataxia than romifidine (higher ataxia scores)? Wouldn't that make romifidine the better choice here? Please explain.

Line 294 - Were you evaluating the number of times the animals had to be repositioned during the MRI? If not, please expand on the subjective findings during the scan. Also, this statement seems to contradict the above statement about romifidine having ataxia scores between 0 and 1 while detomidine scores were all 1. Wouldn't a 0 ataxia horse be less likely to need repositioning?

Line 298 - degrees C I am assuming?

Line 313 - As noted earlier, if you didn't evaluate image quality in some standardized way, you can't say this.

Comments on the Quality of English Language

The manuscript is in places difficult to read or understand, possibly due to translation issues. More significantly, a sentence is not a paragraph - please go through and group similar ideas together into paragraphs or expand on the many isolated sentences written in paragraph form.

Author Response

Dear Reviewer,

thank you very much for your suitable and constructive comments. As suggested by you, we tried to improve the quality of the manuscript on these bases, attempting to bridge the limitations you observed.

Best regards

Cecilia Vullo and co-authors

Comments and Suggestions for Authors

The authors present an interesting manuscript that, while not particularly novel, could have some relevance as standing MRI units become more widespread in practice. I am afraid that I don't really understand the presentation of the data or the conclusions, however.

In the introduction, the authors state that they want to compare 2 CRI protocols, then in the methods they state that a variable rate infusion was used (with a range of doses), then nothing more is said about the infusions - what was the final dosing?, how much total was needed? - was one adjusted more than the other? - etc. It seems that there is a lot of information missing on this aspect of the research.

I completely agree with you. I tried to clarify by adding that the infusion was started with detomidine (0.005 mg/kg/h) or romifidine (0.01 mg/kg/h) and was adjusted assessing depth of sedation and degree of ataxia without exceeding 0.01 mg/kg/h and 0.02 mg/kg/h of detomidine or romifidine respectively. Moreover, I specified between line 231 and 238 that 5 animals of Group R received a supplementary bolus of romifidine (0.01 mg/kg) because at 30 minutes (2 animals), 40 minutes (2 animals) and 80 minutes (1 animal) respectively showed a score 1 of depth of sedation and a score 0 of ataxia, despite the CRI of romifidine was increased to 0.01 mg/kg/h. All 8 animals of Group D showed depth sedation score up to 1, and ataxia score up to 0, as shown in figure 6, and therefore no horses in Group D required an increase of CRI or additional sedation.

In the conclusion, the authors state that detomidine was likely a better choice than romifidine, but this is based on data (better images) that were not actually evaluated. Based on what you have presented - all the exams were completed, the ataxia and sedation levels were not different, etc. - it seems like both drugs are quite equal. If the authors feel like detomidine presents an advantage in this setting as far as image quality, I would suggest having a few radiologists look at all the images and score them based on some quality rubric and then compare that between groups.

Thank you for the comment. We can provide the images, but in any case, I gave more emphasis to the conclusions by clarifying that the animals in group D received the lowest infusion eliminating the reference to image quality.

Other specific comments:

Can the title be a bit more specific - what exactly were you evaluating (not just the drugs used)? Also there seems to be an extra 'a' in the title

I tried to change the title to Comparison of detomidine or romifidine in combination with morphine for standing magnetic resonance imaging in horses

Line 53 - What does the incidence of complications in horses undergoing GA have to do with your study? If you want to say that GA is associated with more complications that standing sedation for MRI, please explain this better.

Thank you. I modified as suggested.

Line 70 - Is there really a need for complete immobility during ultrasound (and perhaps x-ray) examinations? The ultrasound probe tends to move with the patient, does it not?

I’m agree with you. I modified as suggested.

Line 73 - I'm not sure what multimodal analgesia has to do with standing MRI procedures? Do you need analgesia for non-invasive imaging?

I changed multimodal analgesia with neuroleptoanalgesia

Line 93 - As noted above, your entire aim was to compare a CRI of the 2 drugs, but there is very little mention of CRIs in the rest of the manuscript.

Ok, I tried to give more information about the CRI

Line 97 - Your hypothesis has nothing to do with your aim, and wasn't even evaluated in the study. Since every horse received morphine, how could you evaluate whether the 'addition of morphine' did anything? Some of the horses would have needed to not receive morphine in order to evaluate whether the addition of it was beneficial or not.

Ok, I deleted the period.

Line 106 - Does it matter to your results what brand of MRI it was? Please remove if it doesn't.

Ok, done. (…..often if the brand is not reported it is then requested....).

Line 108 - How did you evaluate the patients for 'impaired respiratory or liver function'?

By the horse's history, signs, physical examination findings, and liver-specific enzymes. I modified liver function with liver disorders.

Line 116 - What is a 'bottom of lidocaine'?

Sorry, I changed with button.

Line 134 - Was the infusion dose of drugs varied during the procedure? Or was the infusion rate constant, but different for each horse? Either was, the dosing needs a little more explanation.

Thank you for the suggestion. I tried to explain better the infusion modality.

Line 136 - Your math on the concentrations here seems a little off - if you put 25mg of detomidine in 500mL of saline, the concentration is 0.05mg/mL, not 0.8mg/mL.

Sorry, that's right, I corrected the mistake.

Line 150 - Since you performed multiple comparisons on individual patients, you should include the effect of the individual horse as a random factor in your modeling - and I'm pretty sure that Prism cannot handle random factors in this way. Please check this with your statistician and note how the effect of the individual horse was handled.

The authors agree with the reviewer that Prism is not very versatile regarding the construction of statistical models, however it allows to run a mixed-effect model considering the subjects as random factors. The variation among subjects is highlighted by calculating the SD and Variance between participants. It is also possible to test if the matching is effective and report a p-value that comes from a chi-square statistic that is computed by comparing the fit of the full mixed effects model to the standard model. With a low p-value (as in our study) it is possible to conclude that the matching was effective and the design appropriate.

The Repeated Measure ANOVA was redone fitting a mixed-effect model, where the individual horse was considered as random factor. The main results deriving from the application of the mixed model did not change for HR, RR, RT. The text of the result section, as well as the main RM-ANOVA test results, reports the intrasubject Variance and the results of the test for effectiveness of matching (qui-square data were not reported in the text). The intrasubject Variance was always very low and the matching was effective (low p-values).

Line 170 - You explain the reason for additional sedation in the detomidine group, please do this for the romifidine group or simply note that 2 and 4 horses needed additional sedation in their respective groups.

I apologize for not understanding. I thought I explained the reasons for additional sedation for both groups….

Figures 1 & 5 are redundant to what you noted in the text. Please remove.

Thank you for the comments. Figure 1 was removed. Although the results of Figure 5 may have been described in the text, the authors believe it is appropriate not to remove this figure, to highlight the difference between the two groups regarding the need for additional sedation during MRI.

In Figure 4, you display mean and SEM, but since these scales are ordinal you  should probably show median and range (none of the horses scored a sedation of 1.8...).

Figure 4 has been modified to show the data in median and range. The authors agree with the Reviewer: the appropriate measure of central tendency for ordinal data is the median and range or IQRrange. The only reason why in the previous version of the manuscript the data had been described with mean and sem was exclusively “aesthetic”: since the medians of the two groups were substantially superimposable, the two groups would have been overlapping. In this version we have inserted the median and range, but we have avoided the overlap by slightly nudging the R group to the right (X axis) and up (Y axis), compared to the D group. We think that this graphic stratagem can be a fair compromise between scientificity and aesthetics in reporting data.

Line 195 - So the slope of the line was not zero, were there any differences within groups when comparing different time points? eg. was the RT at 90 minutes significantly different than that at 0 minutes? Also, in the graph your baseline RT (and other physiologic variables) is not shown - you should add this.

Thank you, baseline data for HR, RR and RT were added in the graph.

An additional longitudinal analysis within each group was performed for the three parameters also considering the additional time-point of baseline values.

For HR and RR a significant decrease was observed when comparing the baseline value and each time-point during the MRI. While, as already described in the previous version of the manuscript, during the MRI the HR and RR frequencies remained rather constant without showing any significant longitudinal difference within each group, nor between groups at each time point.

For RT, a decrease in rectal temperature was shown between baseline values and each of the time points during MRI, within each of the two groups. As already described in the previous version, this longitudinal lowering of RT within each group continued to occur in a progressive and significant trend during the MRI, but in any case without showing significant cross-sectional differences between the two groups. The authors believe that reporting all post-hoc significant differences in RT between time-points within each group can be considered redundant, given that almost all comparisons (except immediately or almost immediately subsequent times) were significant and the trend in RT decline was rather linear. Furthermore, in this case the difference is statistically significant but not so relevant from a clinical point of view (from baseline RT to the lowest value, found at 90 minutes, we have a decrease of 0.75 °C in Group D and 0.57 °C in Group R).

Line 209 - If all animals in group D had a sedation score of 'up to 1' (meaning less than 1), shouldn't they all have received additional boluses of sedation?

I’m sorry, I I meant equal to 1

Figure 6 could probably just be noted in the text as average duration and SD or SEM.

Thank you for your suggestion. MRI duration data have been inserted into the text and the figure deleted.

How did you know bowel motility resumed 20 minutes after the end of the exam when you were only evaluating every 30 minutes? Also, what does the 'slowly' modifier mean here?

Sorry, it was a typo. I corrected 30 minutes.

I modified as suggested: The first intestinal auscultation performed 30 minutes after the end of the MRI examination allowed to hear gut sounds.

Line 257 - What exactly were you evaluating that might indicate behavioral changes?

I added that none of the side effects mentioned above occurred

Line 282 - So detomidine causes more ataxia than romifidine (higher ataxia scores)? Wouldn't that make romifidine the better choice here? Please explain.

Absolutely right. The ataxia score was lower in Group R. I than changed the affirmation below where I changed ataxia with depth of sedation.

Line 294 - Were you evaluating the number of times the animals had to be repositioned during the MRI? If not, please expand on the subjective findings during the scan.

Also, this statement seems to contradict the above statement about romifidine having ataxia scores between 0 and 1 while detomidine scores were all 1. Wouldn't a 0 ataxia horse be less likely to need repositioning?

Correct, I tried to modified the phase: Statistical difference in the duration of MRI between groups is related to depth of sedation showed by four subjects in Group R. In fact, the scan was stopped twice in three subjects and three times in one subject to correctly re-place the animals again.

Line 298 - degrees C I am assuming?

Yes, I modified.

Line 313 - As noted earlier, if you didn't evaluate image quality in some standardized way, you can't say this.

Ok, I eliminated the period related the quality of images.

Comments on the Quality of English Language

The manuscript is in places difficult to read or understand, possibly due to translation issues. More significantly, a sentence is not a paragraph - please go through and group similar ideas together into paragraphs or expand on the many isolated sentences written in paragraph form.

We provided to send the paper to improve English editing.

Reviewer 3 Report

Comments and Suggestions for Authors

Author Response

Dear Reviewer,

thank you very much for your suitable and constructive comments. As suggested by you, we tried to improve the quality of the manuscript on these bases, attempting to bridge the limitations you observed.

Best regards

Cecilia Vullo and co-authors

Dear authors,
Thank you very much for your manuscript about Romifidine vs. Detomidine sedation (+ morphine) in horses undergoing MRI.

The study is of clinical importance as both drug combinations are regular used for standing procedures. However, the manuscript itself need rewriting as the focus need to be on the 2 drug combinations rather than on the advantage of standing MRI. Also, an in depth comparison to published sedation protocols is necessary. The discussion needs to be rewritten and also more limitations of the study needs to be highlighted. The lack of assessment of statistical power compromises any result.

Please rephrase Introduction – a short comprehensive note about advantage sedation of general anaesthesia for MRI (less risks associated), than focus on challenge with achieving appropriate level of sedation (no movement etc, highlight no pain stimuli, but maybe weightbearing of painful leg) – short introduction to published protocols (minimize reference of books, as there are some original studies out, rather than having the authors opinion (alternatively discuss what is clinically done by surgeons/anaesthestist preferences, but comment on only handful of studies).

I tried to change as suggested

Also comment on why protocols might differ (eg. why one drug might be advantageous over the other) as you name side effects associated with alpha2s, but not that there might a difference between drugs

I added some study to compare the two drugs, that I commented in the discussion.

As you letter on mention in your aims – that you want to have minimal cardiorespiratory effects, It might be good to highlight/mention associated cardiorespiratory side effects of sedation

Thank you. I added that alfa2 ag decreased respiratory rate

Line 81: are there any papers for advantages of bolus vs. CRI? (there is definitely one recent for dogs Costa et al. 2023)

I’m sorry, but I didn’t find this reference. Could you suggest me the reference?

(Is this? Costa et al. 2023: Effect on physiological parameters and anaesthetic dose requirement of isoflurane when tramadol given as a continuous rate infusion vs a single intravenous bolus injection during ovariohysterectomy in dogs????)

Line 63 and following lines: Alfa2 – replace by alpha2

Sorry, I replace with alpha in all manuscript

M and Ms:
Line 120: how was the anaesthetist unaware of sedation used, when he was the one who gave the drug? Please specify.

Thank you for the comment. I specified that the same anaesthetist evaluating the animals was blinded to the treatment administered that was prepared by an assistant.

Where was sedation (bolus administered) – in stables or already in MRI room?

Ok, I specified as suggested

Line 127: how long after initial bolus administration

I added after 10 minutes

How was HR and RR and T assessed during the MRI?

I added HR (by palpation of the mandibular pulse), RR (by observing thoracic excursion) , RT (using a digital thermometer)

Table 1 and table 2: edit as line breaks are mixed up in my pdf version

Okay, I'll report it. The disposition was modified from the pdf version received from Vet Sci

Line 130: please specify urinary catheter placement: in all animals (female and male?)

Thank you. I added that a urinary catheter provided of a urine collection system was insert in the female horses while a bucket was placed near the penis in male horses

Line 134: as your starting CRI is a range rather than a fixed rate – how did you choose the initial dosing rate?

It was a mistake. I tried to clarify that after 5 minutes an infusion of detomidine (0.005 mg/kg/h) or romifidine (0.01 mg/kg/h) was started using an infusion pump (B Braun, Vista Basic Infusion Pump) that was adjusted up or down assessing depth of sedation and degree of ataxia without exceeding 0.01 mg/kg/h and 0.02 mg/kg/h of detomidine or romifidine respectively.

Line 145: please specify how git motility was assessed

Thank you. I added: Evaluation of cecal motility was evaluated by auscultation at the end of the diagnostic procedure and every 30 minutes for the next 6 hours, using a score of 0-3.

How have initial doses of romidifine and detomidine been chosen? Are these doses considered to be equipotent?

Thank you for the comment. We chose two doses considered equipotent (from Rohrbach et al., 2009, ref 21). I added this consideration in the discussion.

Statistics: Power analysis missing

Power and sample size analyzes were not performed in this study. This was clearly and honestly described (it was further emphasized in this version of the manuscript) as a limitation of the study in the final part of the Discussion section. The absence of an a priori sample size calculation certainly limits the inferential potential of the study and does not allow the results to be interpreted as definitive. The data from the present study are therefore to be considered as preliminary, however they can represent a solid basis from which further more-powered studies could be built (e.g., calculation of the appropriate effect size in order to define the proper sample size to plan an inferential RCCT) with the same design but with greater power and sample size.

Results:
Table 3: why giving t numbers in a simple baseline comparison where p-value would be sufficient? Also – as no difference expect in baseline measurements, mentioning in text would be enough and delete table.

As recommended by the Reviewer, the t data has been eliminated to facilitate the reader's task. Baseline data for HR, RR and RT were reported in the text. The Table 3 was eliminated.

What is about signalement between groups? (breed, sex, body weight?)

Signalment data and statistics were reported at the beginning of the Results section.

Line 169: how do you define suitable or adequate sedation (line 173)? Please clarify and specify in m and Ms?

Ok, I tried to clarified.

Has been temperament included in pre-procedure assessment?

No, unfortunately not.

Duration MRI – please mention differences in text, is not necessary relevant

Ok, I modified as suggested

Please also specify if one or two legs were investigated (it might make a difference if coils are changed

Ok, I specified that in all animals only one leg was investigated

Please give more details to urine production (what do you mean by increased?), how measured?

We did not measure the amount of urine, so we remove the sentence

More details about bowel movement please

Ok, I tried to clarify as suggested

Figure 3: please include parameter on the y-axis (and not only the unit)

As recommended by the Reviewer, the parameters HR, RR, RT were included on the y-axis, in addition to the unit of measurement

Discussion
Benefits and considerations for standing MRI already should have been mentioned in introduction. No need to repeat here as not part of study. A statement – both sedation protocols were feasible for the planned MRI investigations

Ok, done

Line 235: consider listing a few examples rather than the blank statement (which has no content by itself)

Ok I modified as suggested

Line 242: please comment on infra-additive effect of opioids and alpha2 agonists? (Gozalo- Marcilla 2019)

Ok I modified as suggested

Line 244: I assume the depth of sedation and degree of ataxia are the main focus with vital parameters alongside – I would rephrase to make sure, what is your main interest

Ok, I modified as suggested

No discussion why or why not differences between protocols

Paragraphs change between rectal temperature back to ataxia back to rectal temperature

I edited as requested

Please discuss your methods and your results in comparison to relevant literature. Focus on Detomidine vs. romfidine (discuss doses, administration, CRI, timings, etc) also discuss bias assessment etc

Ok, done. I hope to better clarify the discussion.

Why use of opioids at all if no painful procedure

I added that Opioids are often used with α2 agonists to increase clinical effects and minimize side effects, compared to the single use of the drugs [reference]

Discuss use of Acepromazine – other effects

Ok, done

Please also include Freeman et al. 2000 (comparison of romifidine and detomidine)

Thank you, done.

Round 2

Reviewer 3 Report

Comments and Suggestions for Authors

please see my report attached.

Comments on the Quality of English Language

I can see that it has been corrected at the first version, but some minor english editing of the revised version seems necessary. Some sentence formations do seem unfamiliar to me, also use of some phrases/verbs.

Author Response

Dear Reviewer, thank you for your comment, but I would like to inform you that I have already submitted the manuscript for language review (MDPI English editing).
I will send the final version after English editing.

Best regards

Cecilia Vullo

Round 3

Reviewer 3 Report

Comments and Suggestions for Authors

Dear authors,

i am really sorry but it seems like my last review report (the attachment i mentioned in my comments) did not go through.

Would you please be able to review your manuscript accordingly.

I apologize for the confusion as my last decision on V2 (major) must have seem inappropriate considering no further comments. 

Comments on the Quality of English Language

as mentioned in earlier report and also mentioned by authors - final version need some proof reading of english language

Author Response

Dear Reviewer,

thank you very much for taking the time to review our manuscript again. We hope we have clarified what required by following the changes suggested. Thank you for your contributions that allowed us to improve the paper.

Best regards

The author’s

Dear authors,
Thank you very much for the revision of your manuscript about Detomidine or romifidine sedation for standing MRI in horses.
I really appreciate that you have taken all my comments into account.

There are still some flaws with consistency in reporting the study and results.

A major error is the discrepancy in animal numbers in the study (n = 18 in abstract vs n = 16 in the main text). Also, the results section is overcomplicated for this simple and basic study. Your main outcome is – both protocols are feasible without major cardiorespiratory side effect with romifidine requiring more top ups (in the chosen dose range). That is all (and it is important), but should be presented as basic as it is.

Thank you for your comment. I modified the number of animals in the abstract (it was a typo) and added your considerations in the conclusion to highlight the differences between the two groups.

I am also slightly confused by the inconsistency of the reporting (eg use of sem or sd by random for various results).

Sorry for having incorrectly reported SD in the description at times. In any case, the numerical data reported the SEM correctly (we checked all the values several times). In the current version of the manuscript, when the mean is reported as the central tendency index, we have always reported the SEM as the dispersion index (cardinal variables), while when the median is the central tendency index, we have reported the range (min-max) as a dispersion index (ordinal variables). This is now reported more clearly also in the statistical analysis method. Thank you for your comment.

I think an in-depth proof-reading is mandatory before sending back any revision.

Ok, I carried out the suggestion

 Also, I think the language needs to be rechecked after correction of the manuscript.

I have already submitted the manuscript for English editing to the site suggested by Veterinary Science journal.

The references should focus on the context of the study (sedation in horses with different alpha2-agonists), not on feasibility of MRI. It is ok to add a general equine MRI book/reference where sedation (and its challenges) once, but not repeatedly. Also try to minimize use of proceedings, if not really relevant.

Ok, I tried to follow your suggestion

The focus of the study should be introduced and discussed accordingly. Eg. chosen ranges, use of scales, blinded? Observer, algorithm to adjust level etc etc. The discussion of your methods – relating chosen doses and drugs to relevant literature needs to be improved. I suggest a major revision of the discussion as it is very hard to follow with also some errors in referencing studies (at least in the context reported).

I tried to improve the discussion and correct the mistakes regarding the references.

I do have some additional comments.

Introduction:
No need to discuss the advantages of standing MRI over sedation. That is not part of your study. Remove reference 2 and 3

Thank you for your comment, I modified as suggested.

When introducing standing sedation techniques for imaging – I would add also that MRI is focusing on the limbs. However, I do not think that locoregional techniques (relating references) are relevant to your topic/paper and therefore this part can be removed (line 73/74). I would also add, that the challenge is the long procedure of the MRI with acquisition times around 90 minutes.

Ok, I modified as suggested.

Line 76: recommended to be used by CRI over bolus – I think clarification is needed here, that this is to achieve prolonged periods of chemical restraint (as required in MRI), and not as a one of bolus. Furthermore – the references mentioned here – do they really compare bolus vs. CRI? The earlier reference I mentioned, was about tramadol in dogs (CRI vs bolus) , which could be a starting point. There is a recent study by Shane et al. 2021 comparing PK and PD of dexmedetomidine in horses (which I suggest is of interest if you want to keep the statement about CRI better than bolus). However, I am not sure, whether this statement is really relevant or whether you only mention, that it is routinely given as CRI for prolonged chemical restraint (rather than focusing of the advantage of bolus vs CRI)

Ok, I tried to modified as suggested.

Line 89-94: as both groups in your study do receive morphine, I would not go into depth of the introduction with morphine. I would suggest to move up some comments about morphine or opioids, where you mention neuroleptanalgesia/sedation can be achieved by a combination of drugs

Ok, I modified as suggested

Line 88: as you mention correctly, that there are studies comparing romifidine and detomidine – I think you should highlight, the difference in your study by transition into the aim of the study (eg.: “many studies compared these two drugs.....and sedatives. However, to the authors knowledge, the feasibility of either of these protocols for standing sedation for MRI have not been investigated yet. Therefore, the aim of this study was to determine, whether detomidine or romifidine CRI (combined with a single bolus of morphine) provide adequate sedation and immobility with minimal cardiorespiratory effects in horses undergoing standing MRI.

Ok, I modified as suggested.

Materials and Methods
Line 109: please confirm whether it was informed owner consent

Ok, done

Please check your animal numbers... you have 16 here, but mention 18 in the abstract

Ok, I corrected the mistake.

Line 111: “with eater ad libitum” – please change to water ad libitum

Ok, done

Line 119: please include some more details about the blinding (eg. a romifidine bolus for a 500 kg horse works out as 2 ml, whereas detomidine would only be 0.5 ml). how did you ensure that the investigator did not notice the difference?

Ok, I specified that an assistant prepared and injected the bolus.

Same in line 139 – how did you ensure the investigator was unaware of the drug combination if both drugs were made to the same concentration, but delivered at different rates? Also add the initials for the anaesthetist doing the assessments.

I tried to better explain the roles of the anesthesist and the assistant by adding their first name initials.

Line 126: please remove comma after “please see”.

Ok, done

I would also suggest to split this sentence into 2 sentences (eg. : sedation was evaluated. Horses, which deemed inadequately sedated before entering received a supplementary...)

I modified as suggested.

Line 135: please include how the infusion rate was adjusted within the mentioned range. Did you had a certain algorithm or was it purely by the decision of the anaesthetist?

Ok, I specified that VB was adjusted up or down following the decision of CV that assessed depth of sedation and degree of ataxia.

Line 145: please but the reference for the score in brackets

Ok, done

Line 146: what was the route of administration of the top up?

Ok, done

Line 151: was only cecal motility assessed or was normal auscultation of gut sounds performed? As I assume was only right, whereas gut motilitiy would be a standard 4 quadrants. Please clarify

I clarified as suggested

Line 151: change to “Assessment” or “Evaluation” of depth of sedation

Ok, done

Line 162: please check your statistic descriptions and your results – eg you mention reporting of mean and SEM in line 159, but do report mean and sd in line 174.

The authors apologize for this error in reporting the description of the data. Please see previous comment on this topic. Cardinal data is now reported consistently with mean and SD, while ordinal data is now reported with median and range.

Line 164: what matching was performed? Unclear what exactly was compared and why mixed-effect model was used? Between group comparison but also within group comparison? Not clear which variables have been assessed with which test? It becomes clearer when reading the results, but the description is rather overwhelming.

In the previous version of the manuscript (in R2) an additional analysis was performed linked to the previously performed analysis (Repeated Measure ANOVA + post-hoc). The aim was simply to evaluate the influence of the individual horse (considered as a random factor in the additional analysis) with a mixed-effect model. The authors agree with the reviewer that this additional analysis does not bring much benefit, and, at the same time, it can create confusion to the reader, distracting the attention from the principal results. The additional analysis, and therefore also the data of the intrasubject random effect variance (reVar) and the results of the effectiveness of matching (matching p-value) were therefore eliminated.

Results:

Please check whether it is really necessary to give a HR or RR with one decimal place.

Thank you for the comment. Decimal values for HR, RR and RT have now been rounded to the first decimal place.

Line 182: what happened to the horse postponed – was initially sedated, did not work, than repeated? Or aborted before even given any sedation? Also, when was the procedure aborted? As it will be definitely different if the horse had already been in the MRI room (eg

familiarization or even more stressed as panic? Not sure how valid data of this horse will be as I do not know the details, but it definitely needs more information and proper discussion

I clarified that the exam in one horse in Group D was postponed to the next day because the animal appeared too nervous when it had already been in the MRI room, probably due to a panic reaction. The examination was performed 3 days later allowing the owner to enter the MRI room in order the animal feel more comfortable.

line 186: please mention in Material and methods which scores are suitable, when introducing the sedation score.

Ok, done.

Also numbers do not make sense: group D: 6/8 horses score 2. Remaining 2 horses needed top up before entering MRI. Additional? 1 horse score 1 and additional? 1 horse with score 0 (or are these the description of the 2 horses, which did require the top up). Please clarify.

Thank you. It was the description of the 2 horses requiring an extra dose before entering the MRI. The text has been modified accordingly with the aim of allowing the reader to understand.

Please also specify, whether one of the horses was the horse mentioned earlier in line 182.

No, the horses were not the one mentioned in line 182.

line 201: Please add “in all animals” (as you do not specify between groups).

Ok, done.

Can you confirm whether morphine was given before or after the initial sedation assessment mentioned in line 124? (as this might make a difference as morphine can be associated with excitatory effects once given IV)

It was specified that morphine was inoculated after sedation was assessed, after the animals entered in MRI, and before starting the continuous infusion of alpha 2. Do you think it is necessary to explain it better?

line 206: “cross-sectional differences” – please specify

In the previous version, the term "cross-sectional" was intended just to indicate that the comparison was transversal between groups, therefore between the two groups at any time-point. Thanks to the reviewer, the authors believe that the expression "between groups" is enough to describe the type of analysis, therefore the term "cross-sectional" has been removed, since it can be considered a repetition.

the whole paragraph is rather complicated with all the reports of reVar and matching. I am not familiar with this reporting, but I assume neither are the readers. Can you try to simplify this part of the results, especially if there is no statistical significant difference. And the graphs nicely represent your findings. Drop in HR and RR and RT, but no difference between groups.

As recommended by the reviewer, the additional analysis has been eliminated (it was the mixed-effect model, with individual horse as a random factor, auxiliary to RM-ANOVA) because it did not bring much benefit to the study and can create confusion to the reader, as previously described. Accordingly, data of the intrasubject random effect variance (reVar) and the results of the effectiveness of matching (matching p-value) were therefore eliminated from the Result section. The authors believe that the description of the Results is now simpler but clearer than the previous version. Thanks for the constructive comment.

Line 211: how do we know that HR was increased due to morphine? As I understood – horses got initial sedation bolus, was assessed 10 min later, got morphine, moved into MRI room, started CRI and was assessed 10 min later.... so not clear when the increase in HR was. Please specify/clarify.

Ok, I specified that the transitory increase in HR in the horse in Group D occurred five minutes after detomidine bolus administration and in Discussion I citated one paper to justify this event.

Figure 2: please check your data reporting. Is it mean and Sd or mean and sem. Please be consistent.

The authors confirm that the indices of central tendency and dispersion reported in Figure 2 are mean and sem.

Line 221: no need to report p value if it is not significant.

The non-significant p-value was eliminated. Thank you.

Figure 3: for the scores, please add maximum score in the description (easier for the reader rather than looking back), very surprised about reporting of categorical data as median and sem (you use median and range earlier on for similar data – initial sedation score, line 197) Not sure if this figure (number 3) is really relevant and representative. Very difficult to distinguish between the groups. Again, no significant difference and the results are overcomplicated.

Thanks for the comment. The minimum-maximum score achievable for both scales (0 to 3) has been added in the description of the figure. The authors apologize for the error in the description where sem erroneously appeared: the values of the scoring systems are described with median and range, respectively as indices of central tendency and dispersion. Now this is also described in the statistical analysis method in the M&M section. The figure 3 highlight the absence of evident differences between groups in both scoring system and it helps to have information on horses that have reached low levels of sedation and ataxia, however the authors recognize that the distinction between the two groups is not immediate. Therefore, the authors decided to color the groups in Figure 3. Consequently, the colors of all the figures have now changed throughout manuscript, keeping colors constant to indicate groups (Group D: red; Group R: blue). The description of the results has been extensively modified with the aim of making it easier for the reader to understand.

Line 231: there is no figure 6

Thanks for the comment. The authors apologize for the error of having left a residue of a previous version, where the figures were present in greater numbers. "as showed in Figure 6" has been replaced by "as shown in Figure 3".

Paragraph line 227: Please clarify as these results are conflicting and do not make sense:
5/8 animals in group R received a supplementary bolus in addition to the CRI because sedation score was 1 and ataxia score was 0.
No horse in group D received a supplementary bolus in addition to the CRI. All horses had A sedation score of 1 and an ataxia score of 0.

Why did horses in group R got a top up and horses in group D did not get a top up, despite having exactly the same sedation and ataxia score?

Sorry, but there was an error in table 2 (which I have now corrected) that caused me to write 1 instead of 2

Also – how was the CRI rate assessed? How long ago was the change when the top up was required? Did you assess= CRI rate? Or maybe total drug dose used?

Unfortunately, in this study neither the CRI rate nor the total drug dose used were evaluate.

I added this limitation in the discussion.

Maybe horses in group R were maintained initially on the lower rate, which potentially might be insufficient. Turning up the rate might not have made an immediate response. In comparison – detomidine might have been run at the high rate in all horses from the beginning, making them more stable. Please comment and clarify.

Thank you for the comment. I added this consideration in discussion.

Line 243: please bring this statement higher up, when you mention that all horses underwent MRI for navicular bursa syndrome.

Ok, done

Line 244: this is different from your description of material and methods (you state earlier, it was performed at the end of the diagnostics and then q 30 minutes). Here you state it was done 30

minutes after the end.

I modified the statement at line 164

I do not understand what you mean with “allowed for hearing gut sounds”.

Ok, I tried to be clearer.

Please include the results of the gut sounds.

Ok, I added that A score of 2 was assigned to all animals at 30 minutes, 1, 3, and 6 hours.

Discussion:
Please restructure the discussion. It is difficult to read as currently it is jumping repeatedly between different topics – morphine doses at the begin, than behavourial effect in the middle, than obstipation at the end – despite morphine not being part of your comparison. Please summarise.

No need to praise MRI in horses – this study is not about feasibility of MRI in horses. Delete the respective references and statement.

Begin of the discussion is a repetition to the introduction – not sure what line 249-252 tells us.

Ok, I deleted as suggested

When discussion the addition of morphine, maybe also need to comment on potential adverse effects (eg some opioids cause more excitement or head bobbing than others). As you do not compare use of morphine or not, I would mainly comment on its potential usefulness, compare the used dosage and why morphine chosen over eg. butorphanol.

Ok, I modified as suggested

Line 268: when giving doses for CRI please report correct units (per hour or per minute etc)

Ok, done

Line 269: a lower dosage was used for greater confidence with this protocol? Please rephrase this sentence. I guess, the lower CRI rate was chosen as the authors are more familiar with the rate used in this study?

Ok, I modified as suggested

Line 236: lower doses used than reported? Maybe that is why more top ups needed in one group? Please discuss this.

Ok, I discuss this point as suggested.

Line 264: please correct your statement. Freeman and England postulate an equipotency of 20 mcg/kg of detomidine and 120 mcg/kg of romifidine with regard to its sedative effects. These drug doses have not been used in the current study (and even if extrapolating from the dose to the lower dose rate, what we actually can’t do – 10 mcg/kg detomidine would still not equal 40 mcg/kg romifidine (as used in this study), but 60 mcg/kg)

Ok, I deleted this statement

Line 276: if we do not need a morphine CRI as MRI is not a painful procedure, why do we give morphine at all? Please comment and clarify. I assume morphine has been given to enhance sedation (and also maybe minor pain relief with an ongoing orthopaedic issue to maintain full weight-bearing throughout the study). However, as the effect of morphine can be 4-8 hours – not top up of morphine (or CRI) is needed. Maybe this should be included rather than commenting on pain relief or not

Ok, I modified as suggested

Line 283: you mention that no side effect has been reported, but in results you stated the increased HR after morphine?

Ok, I added a reference to clarify this collateral effect.

There is no need to extend/increase length of discussion about morphine use or not (again – you do not compare morphine). Just find literature (papers) supporting your dose and usefulness. Mention that some side effects (eg drop in HR, RR etc) seen during the sedation/MRI) might be due to the morphine itself. It might also mask, therefore, any minor difference between detomidine/romifidine

Ok, I deleted the Hammad reference.

Reference 19 used a similar bolus in both groups, but 4 times higher CRI rates. You still mention that you have chosen your protocols from that study.

Concerning morphine bolus it was the same, and we explained that we did not use CRI because there was no surgical stimulation.

Line 327 – you do not know whether any histamine release happened in your horses? you reported one transient tachycardia? Maybe in response to a drop in blood pressure following histamine release? We do not know, so just state it was not investigated in this study (also no BP measurement performed)

Ok, I delated this statement

Obstipation morphine – I assumed the test of GI motilitiy was to asses differences in the two groups – this is not mentioned in the discussion at all. What is about alpha 2 agonists and gut stasis? Please include this.

Indeed, the Reduction in intestinal motility can be caused by morphine.

Limitations of the study – compare you animal numbers to similar papers? Is 8 horses per group normal?

Ok, I added one references as suggested

Adjustment of CRI rate to be discussed

I added this limitation in the discussion.

Used scores to ve discsussed

I didn’t understand

Line 343: why reduce romifidine loading dose but not detomidine, when assuming morphine can help reducing the alpha 2 requirement

In fact, we supposed that the initial dose of romifidine was low

Line 335: I guess, this is a major factor, as this might discourage people from using the respective protocol. I would highlight this fact much earlier. And also add in the results section (when giving the timings), how often repositioning was necessary in both groups.

I tried to modify as suggested

Round 4

Reviewer 3 Report

Comments and Suggestions for Authors

Dear authors,

thank you very much for your careful revision of your mansucript about alpha2 cris for standing MRI in horses.

The paper is now easy to follow and to understand . I appreciate your work on that. 

Here are just a few minor comments:

Line 54: recently – replace with “these days”

Line 77: rephrase –

… most commonly drugs used in horses for standing sedation. For standing procedures requiring long-term chemical restraint as for MRI, a CRI of these drugs is often administered. (4,14, 15)

Line 142: remove “schauvliege et al.” and just keep the reference number

Line 183: please add. “Only the 2nd (successful) attempt was included in analysis for this study”.

Line 252:”that the combination…”

Line 257-260: I would suggest to move this paragraph right at the begin of the discussion., than continue with “several studies…”

Author Response

Dear Reviewer,

thank you for all the suggestions that helped us significantly improve our manuscript.
I made all the corrections but I am not sure I understand where to add the sentence: "Only the 2nd (successful) attempt was included in analysis for this study."

Best regards

Cecilia Vullo
